# Heart Rate Variability Monitoring during a Padel Match

**DOI:** 10.3390/ijerph19063623

**Published:** 2022-03-18

**Authors:** Jose A. Parraca, Joana Alegrete, Santos Villafaina, Nuno Batalha, Juan Pedro Fuentes-García, Diego Muñoz, Orlando Fernandes

**Affiliations:** 1Departamento de Desporto e Saúde, Escola de Saúde e Desenvolvimento Humano, Universidade de Évora, 7004-516 Évora, Portugal; jparraca@uevora.pt (J.A.P.); joana.alegrete@fa.uevora.pt (J.A.); svillafaina@unex.es (S.V.); nmpba@uevora.pt (N.B.); orlandoj@uevora.pt (O.F.); 2Comprehensive Health Research Centre (CHRC), University of Évora, 7004-516 Évora, Portugal; 3Faculty of Sport Sciences, University of Extremadura, Avenida de la Universidad s/n, 10003 Cáceres, Spain; jpfuent@unex.es

**Keywords:** exercise, non-linear dynamics, autonomic nervous system, heart rate monitoring

## Abstract

Padel, an increasingly popular sport, presents some associated cardiovascular risks, which can be prevented by heart rate variability (HRV) monitoring. However, there is no study worldwide that characterizes HRV during padel games. Therefore, this study aims to monitor HRV responses and characterize them using linear and non-linear metrics at three timepoints: at baseline, during a game, and during recovery. Twenty-seven amateur participants had their HRV monitored before, during, and after a 90 min padel game. We extracted time, frequency, and non-linear measures with MATLAB for baseline, rest period, and at three periods of 5 min during the game. The differences in measures were assessed through an ANOVA. The autonomic modulation was affected by the padel match in amateur players. In this regard, the root mean square of successive differences between normal heartbeats (RMSsd), Poincaré plot (perpendicular standard deviation (SD1) and horizontal standard deviation (SD2)), sample entropy (SampEn), low frequencies (LF), and very low frequencies (VLF) were significantly reduced during the game, whereas alpha-2, high frequencies (HF), and the ratio between low and high frequencies (LF/HF) increased during the game. Furthermore, an abrupt change was found in the autonomic modulation between game and recovery assessments, which suggests the need to rethink the practices of cool-down protocols. The multiple timepoints analyzed during this study allow us to investigate the evolution of different HRV measures in the time, frequency, and non-linear domains, clarifying the interpretation of the variables, especially the less-investigated ones such as the non-linear measures.

## 1. Introduction

Padel practice has experienced enormous growth worldwide and has been increasing exponentially and steadily since 2010 [1]. This racket sport is practiced in pairs in a rectangular playing area of 10 by 20 m, divided into two halves by a central net. The padel court is characterized by a completely closed perimeter, combining areas of metal mesh and materials that allow a regular bounce of the ball against the baseline and the sidewalls [2]. These characteristics could categorize padel as a long-lasting intermittent medium-/high-intensity sport [3] with numerous physical, psychological, personal, and social benefits [4,5]. In addition, this sport’s development and expansion have been favored for its ease of technical learning [6], being practiced by four players [4], and its longer point duration, allowing for a longer playing time than other racket sports [7,8].

Similar to other intermittent sports, aerobic and anaerobic metabolic pathways are used during padel practice [9,10], reducing the risk of death associated with cardiovascular diseases by 59% [11]. However, padel has an estimated risk of injury of 2.75 per 1000 h of play, similar to the tennis injury rate [12]. These injuries affect amateur players more frequently [13]; these include specific cardiovascular complications [12].

Heart rate variability (HRV) shows the interactions between the heart, the brain, and the dynamics of two branches of the autonomic nervous system: the sympathetic (SNS) and parasympathetic (PNS) nervous systems [14]. Through the analysis of HRV, it is possible to extract several biomarkers, which are robust indicators of health levels, well-being, and resilience to stress [15,16]. A low HRV reflects a minor adaptive capacity to the environment. Therefore, it is possible to detect and prevent PNS imbalances, mortality, and other diseases [17,18]. In sport, HRV has been studied in athletes and non-athletes during physical training workloads, interventions, during sleep [19], or even after ultra-endurance competition [20] to prevent injuries and optimize training load and performance [21]. Consequently, understanding HRV behavior during a game of padel may help reduce cardiovascular complications and might become one of the most fundamental risk reduction and prevention tools. HRV analysis is calculated through RR interval time extraction [15] using two approaches: linear models and non-linear dynamic analysis methods [22]. Linear analysis includes time-domain analysis and frequency-domain analysis. Time-domain analyses are all correlated because they derive directly from RR intervals. For instance, the root mean square of successive differences between normal heartbeats (RMSSD) [23] might indicate PNS modulation [16], with higher values indicating great variability and, therefore, a lower risk of sudden death [24].

Frequency-domain analyses, as the name implies, are spectral analyses [23], which take into account three frequencies of HRV [16]. High frequencies (HF), located between 0.15 and 0.4 Hz, reflect parasympathetic vagal activity corresponding to cardiac variations related to the respiratory cycle [24], which decrease during stress, panic, and anxiety situations [24]. Low frequencies (LF), located between 0.04 and 0.15 Hz, reflect PNS and SNS functions, changes in blood pressure, and efferent respiratory functions [24]. Finally, very low frequencies (VLF), located between 0.0033 and 0.04 Hz, are associated with thermoregulation, hormone levels (testosterone), and cardiac endothelial influences [24]. The low and high frequencies ratio (LF/HF ratio) is still poorly understood, but it conveys the PNS domain over the SNS [24].

Non-linear analysis emerges from heartbeat time series observations, which, despite being random, follow fractal patterns. These time series have similarities with time scales that obey 1/f fluctuations, such as pink noise [25], a pattern that has served as a model for many biological systems and psychological states [26]. We can calculate two metrics through a Detrended Fluctuation Analysis (DFA): Alpha-1, an indicator of short-term self-similarity, between 4 and 16 heartbeats; and Alpha-2, a long-term indicator, between 16 and 64 heartbeats [27]. Two other indicators extracted from the non-linear analysis are the standard deviation of the distribution of the Poincaré plot, the perpendicular standard deviation (SD1) that characterizes the PNS modulation, and the horizontal standard deviation (SD2), which represents the modulation of the SNS and PNS in long-term monitoring [15]. Another widely used indicator is sample entropy (SampEn), a negative logarithm of the conditional probability of two similar sequences [28]. Low values characterize regular time series, whereas higher values show more complex systems and are associated with healthy behaviors [27].

Although the study of HRV is a valuable method to understand the role of the ANS modulation in body regulation and cardiovascular control, to the best of our knowledge, no studies have evaluated HRV during padel practice. A previous study evaluated heart rate (HR) in 24 players during padel games, showing intensities between 50% and 70% of the Maximum Heart Rate (HRmax) [3]. Furthermore, the acute effect of a paddle game (1 h 30 m—three sets) was verified at the HR level, in four amateur players [10], and the authors showed that the game averages were around 125 beats per minute, with an average effort of 70% of HRmax, similar to the results of those obtained by other authors [29]. Although these studies have analyzed the cardiorespiratory function during padel practice [10], there are scientific gaps in the analysis of HRV using linear and non-linear metrics. Furthermore, as previously reported, HRV monitoring can prevent these sport-related specific cardiovascular complications [12].

Our study seeks to be the first in padel research to monitor HRV responses and characterize them using linear and non-linear metrics at three timepoints. Furthermore, it also aims to document changes in autonomic modulation during these different timepoints (rest, game, and recovery). Therefore, the results would allow us to characterize autonomic modulation during intermittent exercise. These results would be useful for physical trainers and researchers. Regarding physical trainers, we will provide a characterization of HRV during exercise and recovery. Thus, they may use this protocol to assess the fatigue of their athletes, knowing when they can continue applying training loads. Furthermore, this study would be of interest for researchers in the field of HRV. In this regard, there is controversy regarding the interpretation of some HRV indexes, specifically those related to frequency-domain and non-linear measures [24]. Thus, this paper would help future studies to interpret HRV data as well as HRV behavior during and after exercising.

## 2. Materials and Methods

### 2.1. Design and Subjects

A total of 27 amateur padel players (aged 37.26 ± 9.42) participated in this cross-sectional study (Table 1). The study was carried out in indoor paddle facilities from April 2019 to June 2019. Participants’ inclusion criteria were: (1) to practice padel between four and six hours/week; (2) to be classified by a padel expert as of medium level; and (3) to be aged between 18 and 65 years old.

Participants were excluded if they: (1) had a condition that might make padel contraindicated, such as retinopathy, musculoskeletal injuries, or major balance problems; or (2) had taken any substance or drug that affects the autonomic nervous system 48 h before starting the protocol.

### 2.2. Ethical Considerations

All the procedures were approved by the University research ethics committee (approval number: 89/2018) and followed all ethical guidelines under the European Standard of Good Clinical Practice (ICH-GCP Guidelines) and the Declaration of Helsinki. Participants were informed about all the procedures, which were described in the informed consent. In addition, they were asked to read and sign the informed consent. A signed copy of the informed consent was given to the participants.

### 2.3. Procedures

A technician randomly allocated the twenty-seven participants in seven groups of four. Then, couples were randomly configured. The technician did not take part in the evaluation or data acquisition. Another researcher developed both the assessment and the data analyses. This researcher was blinded to the grouping allocation.

Regarding assessments procedures, participants had to be at rest for 15 min prior to collection of baseline HRV in a sitting position, as recommended by Catai, Pastre, de Godoy, da Silva, de Medeiros, Takahashi, and Vanderlei [15] and the European Society of Cardiology and the North American Society of Pacing and Electrophysiology [30]. After 15 min at rest, 5 min of baseline was collected. Subsequently, participants performed ten minutes (min) of warm-up in the padel court. Then, they had to play a 90 min padel match while HRV was recorded. Immediately after the game, participants completed five minutes of HRV recording at rest (in the same conditions as the baseline assessment).

Three periods of 5 min each were extracted during the 90 min of the game: one in the first 30 min, another between 30 and 60 min, and finally, between 60 and 90 min of the game. This procedure allowed us to study the HRV at three different game timepoints. HRV measurements take five minutes, since it is considered the gold-standard for short-term measurements [24], showing an ICC = 0.97 (0.81–0.99) [31] for HRV variables such as RMSsd. 

### 2.4. Materials

The HRV was acquired using a H10 chest strap (Polar Inc., Kempele, Finland) and recorded using a RS800CX monitor (Polar Inc., Kempele, Finland). This wireless device was placed below the participant’s chest muscles, allowing a reliable recording [32]. 

### 2.5. Data Analysis

The Kubios HRV software (v. 3.3) [33] was used to pre-process and obtain HRV data. A median filter was applied to correct possible artefacts. This filter allows the identification of RR intervals shorter/longer than 0.25 s, compared to the average of the previous beats. Correction replaces the identified artifacts with cubic spline interpolation. All HRV indices were extracted using MATLAB Release 2019a (The MathWorks, Inc., Natick, MA, USA). Time-domain, frequency-domain, and non-linear measures were extracted.

The following metrics were calculated: Time-Domain Analysis: (a) square root of differences between adjacent RR intervals (RMSsd).Frequency-Domain Analysis: (b) spectral analysis of the very low frequency (VLF, 0.00333–0.04 Hz), low frequency (LF, 0.04–0.15 Hz), and high frequency (HF, 0.15–0.4 Hz) ratio (LF/HF).Non-linear analyses: (c) non-linear metrics: the RR variability from heartbeat to short term Poincaré graph (width) (SD1), the RR variability from heartbeat to long-term Poincaré graph (length) (SD2), short-term fluctuation of the detrended fluctuation analysis (alpha-1), long-term fluctuation of the detrended fluctuation analysis (alpha-2), and the sample entropy (SampEn), which measures the regularity and complexity of a time series.

### 2.6. Statistical Analysis

Taking into account the results of normality (Shapiro–Wilk) and homogeneity, parametric statistics were employed. Thus, an analysis of variance for repeated measures (ANOVA) was performed to assess the differences between variables for each timepoint. Moreover, *t*-tests for paired samples were conducted to perform post hoc analyses. *p*-values were corrected using the Bonferroni correction for multiple comparisons in order to minimize Type I error. The partial eta squared (η2p) effect size was calculated. The significance level was set at 0.05. The Statistical Package for Social Sciences software version 24.0 (IBM SPSS Statistics for Windows, IBM Corp., Armonk, NY, USA) was employed to conduct the statistical analyses.

## 3. Results

Table 2 shows the descriptive analysis for all HRV data across timepoints. Regarding HRV, repeated measures ANOVA showed significant differences for all studied variables (*p* < 0.001) (see Table 2).

Complementarily, Figure 1 depicts the evolution of HRV at baseline, after 30 min of the game, after 60 min of the game, after 90 min of the game, and during recovery. Regarding time-domain variables, RMSsd showed a significant decrease between baseline and the rest of the timepoints. Furthermore, the RMSsd value during recovery was higher than at the three game timepoints (at 30, 60, and 90 min) (see Figure 1 for details). 

Frequency-domain variables showed significant differences. In this regard, HF showed a significant decrease between baseline and the rest of the timepoints. In the LF, values significantly increased during recovery compared with game timepoints (30, 60, and 90 min). The VLF showed a significant decrease in the three analyzed timepoints compared to the baseline, as well as a significant increase in these variables when compared to recovery. A significant increase in the LF/HF ratio was observed between baseline and the first timepoint (at 30 min), and a significant increase in recovery was observed compared to baseline, 60 min, and 90 min assessments.

Figure 1 also depicts the changes in non-linear measures. A significant increase in alpha-1 was observed in recovery, compared with the rest of the assessments. In alpha-2, a significant increase was observed between baseline and the rest of the assessments, and a significant decrease was observed when comparing recovery to the game assessments (at 30, 60 and 90 min). In SampEn, a gradual reduction can be observed during the protocol, achieving a significant decrease between baseline and the rest of the assessments. Regarding SD1 and SD2, similar results have been shown, with a significant reduction between baseline and the rest of the assessments and a significant increase after the game, during recovery. 

## 4. Discussion

The objective of this study was to investigate the impact of a padel game on autonomic modulation. The results showed significant changes in the time-domain, frequency-domain and non-linear variables of HRV. The results are interesting since this is the first study investigating autonomic modulation during a padel game with multiple timepoints analyzed. This study allows us to examine the evolution of different HRV measures in time, frequency, and non-linear domains, clarifying the interpretation of the variables, especially the less-investigated ones such as the non-linear measures.

One of the objectives of this study was to observe the evolution of HRV through a padel game. In this regard, previous studies have highlighted the usefulness of RMSSD, a time-domain variable, in assessing autonomic modulation due to the high sensitivity of this variable [34] to parasympathetic modulation [24]. In our study, a U-shaped effect on RMSsd was found. This is in line with previous studies [24,35], with a decrease in RMSsd during exercise and a posterior increase during recovery. Previous studies have examined the impact of different exercise modalities on HRV. Similar findings of RMSsd reduction during exercise have been observed during arm and leg exercises, cycling exercise, running, and incremental treadmill exercise or judo, among others [36,37,38,39,40,41,42,43]. Furthermore, some of these studies also reported a reduction in SD1 [41,42,43]. This is consistent with previous studies that showed the relationship between SD1 and RMSsd.

In the present study, we analyzed HRV data at three timepoints during a padel game: at 30, 60, and 90 min. We aimed, with these three measurements, to ascertain if we were able to observe the effect of fatigue in HRV. However, significant differences between these three timepoints were not found in any of the HRV variables studied. This could indicate that even in an intermittent game of medium/high intensity [3], these changes are not perceived as a stressful physiological stimulus [24]. Since changes in HRV during exercise are intensity-dependent [44], these findings could mean that intensity is maintained (without drastic increments) during the padel game.

Regarding the recovery phase, five minutes at rest, a significant increase was found. This is in line with previous research, where upon exercise cessation, HRV measures show a time-dependent recovery [44]. In this regard, intensity has been proposed as the primary determinant of post-exercise recovery. Another study showed that energy expenditure is a determinant of post-exercise parasympathetic reactivation [45]. Cunha, Midgley, Gonçalves, Soares, and Farinatti [45] showed that the recovery of RMSsd was more rapid after exercises comprising small muscle mass. In [46], acute recovery at different exercise intensities and durations was investigated. The results revealed that the parasympathetic recovery was slower after performing intense and moderate exercise. Nevertheless, RMSsd seemed to be recovered after 40 min in most of the conditions studied. Thus, we hypothesized that five minutes might not be enough to recover initial values of parasympathetic activation. This is interesting because, with systematic control of this outcome, HRV could be used as a guide to achieve better timing in training prescription [47].

Apart from time-domain results, we reported frequency-domain and non-linear variables. Non-linear variables are less studied, despite the relevant information they can provide. A previous study showed that the short-term scaling component (alpha-1) is modified by training with an inverted-U during an incremental test [48]. This bi-phasic response is consistent with the bi-phasic nature of parasympathetic reflex controls of HR as a function of exercise intensity [49]. In our study, an increase in short-term self-similarity (alpha-1) and a reduction in long-term self-similarity (alpha-2) can be observed. This could demonstrate a sudden change in the physiological requirement of the organism, increasing the possibility of cardiovascular risk [27]. This abrupt change evidences the importance of a cool-down phase, suggesting that a similar activity, but with less intensity in terms of heart rate, would be necessary to avoid an accentuated response in short-term self-similarity, obtaining a cardiac effect with a protective character. Regarding frequency-domain variables, significant changes were detected in the frequency domain. However, due to complex sympathetic–parasympathetic interactions, the underlying mechanisms of frequency-domain variables are less well established. Thus, controversial results are available in the literature [44]. Nevertheless, this study reports frequency-domain and non-linear measures in order to increase the available literature regarding this topic. 

This study has some limitations to be acknowledged. First, the study was focused on padel and, specifically, on amateur players (classified by a padel expert to be of medium level). We recommend replicating this study in other sports including elite players to compare the HRV responses among these groups. In this regard, the inclusion of two groups would allow a study of whether the control of the SNA response in the cool-down or in the different game phases statistically differs between these groups. Second, the relatively small sample size does not allow the extrapolation of the results to the general population. Third, the physical fitness of participants was not directly taken into account. Thus, the results could be influenced by this factor. However, this study has some strengths that should also be recognized. We reported the evolution of HRV during a physical exercise, using padel as a model, extracting different variables of time, frequency, and non-linear domains. This is relevant since, in the study of HRV, there is controversy regarding the interpretation of some variables, especially in some of the frequency-domain measures, whereas others are poorly studied, such as the non-linear measures [24]. These results and the graphical representation of them would allow researchers and trainers to interpret their studies, since the evolution of HRV has been monitored under a physical stimulus (padel in this case). Furthermore, we reported that 5 min of recovery was not enough to recover the baseline level of HRV. Therefore, HRV monitoring could be used as a critical outcome in the training and decision-making process. In this regard, systematic assessment with a baseline and post- and pretraining measures could be useful in training prescription. These systematic processes could reduce the risk of overtraining [50].

## 5. Conclusions

The results showed significant changes in the time-domain, frequency-domain, and non-linear variables of HRV when comparing baseline with during game and recovery timepoints. Furthermore, the HRV results indicated a need for behavioral changes in cool-down after an amateur padel game to avoid an abrupt change in autonomic modulation. Therefore, HRV monitoring should be systematically used by coaches or trainers to monitor games and training in order to reduce the risk of injury or overtraining. 

## Figures and Tables

**Figure 1 ijerph-19-03623-f001:**
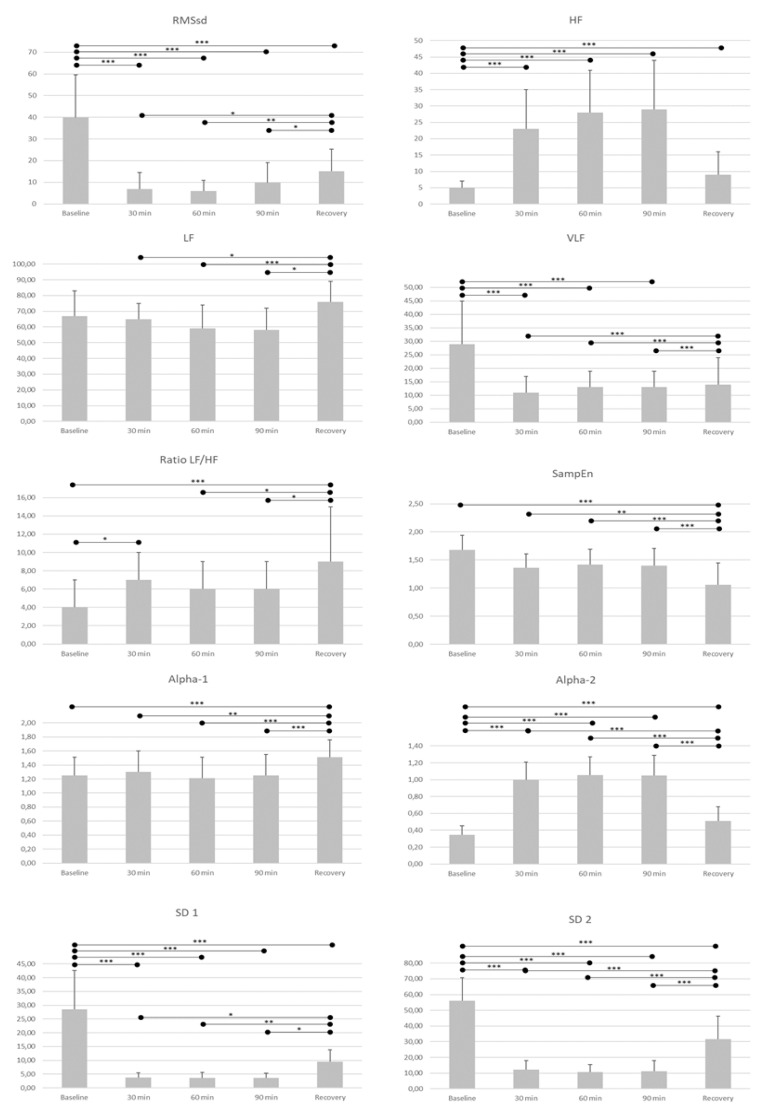
Changes of the time-domain variable (RMSsd—(ms)), frequency-domain variable (HF, LF, VLF—(Hz) and LF/HF ratio), and the non-linear measures (Alpha-1, Alpha-2, SampEn, SD1 and SD2 (ms)), during the protocol: at baseline, after 30 min of the game, after 60 min of the game, after 90 min of the game, and during recovery. *p* < 0.05 *, *p*< 0.01 **, *p* < 0.001 ***. RMSSD: the square root of the mean of the squares of the successive differences of the interval RR; Alpha-1: Short-term fluctuation of the detrended fluctuation analysis; Alpha-2: Long-term fluctuation of the detrended fluctuation analysis; SD1: Dispersion, standard deviation of points perpendicular to the axis of line-of-identity in the Poincaré plot; SD2: Dispersion, standard deviation of points along the axis of line-of-identity in the Poincaré plot; LF: Low frequency; HF: High frequency; VLF: Very low frequency; LF/HF: Low frequency (LF) ratio (ms2)/High frequency (HF) (ms2).

**Table 1 ijerph-19-03623-t001:** Characteristics of participants.

Variables	Mean (SD)
Age	37.26 (9.42)
Height (cm)	175.26 (5.05)
Weight (kg)	80.93 (12.67)
BMI (kg/m^2^)	26.26 (3.21)
% Fat mass	21.41 (5.94)

BMI: Body mass index.

**Table 2 ijerph-19-03623-t002:** Descriptive (mean ± standard deviation) and inferential analysis of HRV during an amateur padel game.

	Baseline	30 min	60 min	90 min	Recovery	F	*p*-Value	η^2^p
RMSsd (ms)	39.87 ± 19.64	6.76 ± 7.69	5.96 ± 4.98	6.84 ± 9.31	15.06 ± 10.15	60.58	<0.001	0.700
Alpha-1	1.25 ± 0.26	1.30 ± 0.30	1.21 ± 0.30	1.25 ± 0.30	1.51 ± 0.25	8.39	<0.001	0.244
Alpha-2	0.34 ± 0.11	1.00 ± 0.21	1.05 ± 0.22	1.05 ± 0.24	0.51 ± 0.17	8.39	<0.001	0.244
SD1 (ms)	28.24 ± 13.91	4.78 ± 5.44	4.22 ± 3.53	4.84 ± 6.59	10.67 ± 7.19	60.57	<0.001	0.700
SD2 (ms)	56.06 ± 14.78	12.15 ± 5.74	10.68 ± 4.90	11.26 ± 6.75	31.70 ± 14.58	145.29	<0.001	0.848
Sample Entropy	1.68 ± 0.26	1.36 ± 0.25	1.42 ± 0.27	1.40 ± 0.31	1.06 ± 0.39	17.13	<0.001	0.397
LF(Hz) (%)	67 ± 16	65 ± 10	59 ± 15	58 ± 14	76 ± 13	9.061	<0.001	0.266
VLF (Hz) (%)	29 ± 16	11 ± 6	13 ± 6	13 ± 6	14 ± 10	20.162	<0.001	0.446
HF (Hz) (%)	5 ± 2	23 ± 12	28 ± 13	29 ± 15	9 ± 7	28.615	<0.001	0.534
Ratio LF/HF	4 ± 3	7 ± 3	6 ± 3	6 ± 3	9 ± 6	6.577	<0.001	0.208

RMSSD: the square root of the mean of the squares of the successive differences of the interval RR; Alpha-1: Short-term fluctuation of the detrended fluctuation analysis; Alpha-2: Long-term fluctuation of the detrended fluctuation analysis; SD1: Dispersion, standard deviation of points perpendicular to the axis of line-of-identity in the Poincaré plot; SD2: Dispersion, standard deviation of points along the axis of line-of-identity in the Poincaré plot; LF: Low frequency; HF: High frequency; VLF: Very low frequency; LF/HF: Low frequency (LF) ratio (ms2)/High frequency (HF) (ms2).

## Data Availability

The data presented in this study are available on request from the corresponding author. The data are not publicly available due to privacy.

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
