# Peer review of "Heart Rate Variability Monitoring during a Padel Match"

_ijerph, 2022, doi:10.3390/ijerph19063623_

Round 1
Reviewer 1 Report
This study seeks to be the first in padel research to monitor HRV responses and characterize them using linear and non-linear metrics at three moments . Furthermore, it also aimed to document changes in the autonomic modulation during these different moments. The study turns out to be well written and, considering the great importance of padel in the last period, it is decidedly current. In addition, measuring heart rate variation over different periods is a good way to assess post-exercise performance and recovery. I believe the manuscript needs minor changes. In particular, for example, the authors could consider in the introduction a work that analyzed the variability of the heart rate before, during and after an ultraendurance swim:
Valenzano et al., Heart-rate changes after an ultraendurance swim from Italy to Albania: A case report, International Journal of Sport Physiology and performance, 2016;
Author Response
We would like to thank the reviewer for their detailed comments and suggestions on the manuscript. They have identified important areas which required improvement. We have carefully addressed all the reviewers’ suggestions and provided a detailed point-by-point response to each comment. Please find below a point-by-point response to reviewer’s comment. Reviewer's comments are in bold and numbered (R1.1, R1.2…). We use red colour to response.
----
REVIEWER #1
R.1.1.- This study seeks to be the first in padel research to monitor HRV responses and characterize them using linear and non-linear metrics at three moments. Furthermore, it also aimed to document changes in the autonomic modulation during these different moments. The study turns out to be well written and, considering the great importance of padel in the last period, it is decidedly current. In addition, measuring heart rate variation over different periods is a good way to assess post-exercise performance and recovery. I believe the manuscript needs minor changes.
Authors: Thank you for your valuable feedback.
R.1.2.- In particular, for example, the authors could consider in the introduction a work that analyzed the variability of the heart rate before, during and after an ultraendurance swim: Valenzano et al., Heart-rate changes after an ultraendurance swim from Italy to Albania: A case report, International Journal of Sport Physiology and performance, 2016;
Authors: Thank you for providing us this interesting and timely reference. We have included it into our introduction.
R.1.3.- Finally, it would be advisable to specify all the acronyms used both in the tables and in the graphs.
Authors: We absolutely agree with your comments. Thus, we have included in the footnotes the acronyms for all the variables.
I would like to sincerely thank for your attention and time devoted to the present work, and we will be very pleased to publish it in your fine journal

Reviewer 2 Report
Your study(manuscript itself) must be improved prior to resubmitting to this quality journal. Your study would be an example of low quality editorial works and mis interpretation of the data compiled. Moreover, I don't find great merits for potential readers of this study. In other words, what are new scientific findings resulted from your study? Here are some detailes of my comments for further steps of article preparation for your study:
- in section 2, A total of 27 amatuer participants then you mentioned 28 participants in the next part of the same section
- page 3-5, I see multiple errors such as pacing issue, multiple period marks (many)
- Informed consent shoud be "informed by the researchers" instead given, read, and singed by themselves - inappropriate explanation regarding this matter
- Reliability and Validity both are scientific but still subjective index as to potential quality of your data collected. It is not a matter of how you can judge as excellent or great. It can only be proved or described by numeric value so consumers of your study judge themselves.
- section 2.4 must be titled differently
- Avoide Type I error? in the scientific world of "inferential statistics" not possible
- V24.0 - You need to spell out "version"
- missing citation and not properly referenced your source
- red highlighted areas? should be avoided prior to resubmitting your manuscript
Author Response
We would like to thank the reviewer for their detailed comments and suggestions on the manuscript. They have identified important areas which required improvement. We have carefully addressed all the reviewers’ suggestions and provided a detailed point-by-point response to each comment. Please find below a point-by-point response to reviewer’s comment. Reviewer's comments are in bold and numbered (R1.1, R1.2…). We use red colour to response.
REVIEWER #2
Thank you for your valuable and constructive feedback. We truly believe that after considering your suggestions, the quality of the manuscript has been significantly increased.
R.2.1.- Your study (manuscript itself) must be improved prior to resubmitting to this quality journal. Your study would be an example of low quality editorial works and mis interpretation of the data compiled. Moreover, I don't find great merits for potential readers of this study. In other words, what are new scientific findings resulted from your study?
Authors: We absolutely agree that the quality of the editorial works was unappropriated. Probably, a human or informatic error occurred previous to submit the manuscript. Therefore, we want to apologize for this mistake.
Moreover, we understand your position regarding the scientific findings of our results. Therefore, we have reinforced the fact that there are controversy regarding HRV indexes interpretation (Shaffer and Ginsber, 2017). Thus, the monitorization of HRV during a stimulus (in this case a 90 min padel match) could be of great help for the interpretation of future studies in the field of HRV (independently if the study will be conducted in the field of neuroscience, physiology or sport science). We have included this fact in the introduction, before objective since we totally agree that the rationale of the study was missing.
“Therefore, results would allow to characterize the autonomic modulation during an intermittent sport. These results would be useful for physical trainers and researchers. Regarding physical trainers, we will provide a characterization of HRV during exercise and recovery. Thus, they may use this protocol to assess the fatigue of their athletes, knowing when they can continue applying training loads. Furthermore, this study would be of interest for researchers in the field of HRV. In this regard, there is controversy regarding the interpretation of some HRV indexes, specifically those related to frequency domain and non-linear measures [24]. Thus, this paper would help future studies to interpret HRV data as well as HRV behavior during and after exercising”.
Shaffer, F.; Ginsberg, J.P. An overview of heart rate variability metrics and norms. Frontiers in public health 2017, 5, 258.
R.2.2.- In section 2, A total of 27 amateur participants then you mentioned 28 participants in the next part of the same section
Authors: Thank you for pointing us this typo. It has been corrected.
R.2.3.- Page 3-5, I see multiple errors such as pacing issue, multiple period marks (many)
Authors: Thank you for pointing us this issue. We have rewritten some paragraph and sentences in order to solve this problem.
R.2.4.- Informed consent shoud be "informed by the researchers" instead given, read, and singed by themselves - inappropriate explanation regarding this matter
Authors: We see your points. We have modified the statement for better clarity.
“Participants were informed about all the procedures, which were described in the informed consent. Also, they were asked to read the informed consent and signed it. A signed copy of the informed consent was given to the participants”.
R.2.5.- Reliability and Validity both are scientific but still subjective index as to potential quality of your data collected. It is not a matter of how you can judge as excellent or great. It can only be proved or described by numeric value so consumers of your study judge themselves
Authors: Thank you for your kind recommendation. We have modified the statement in order to avoid personal judgments.
R.2.6.- Section 2.4 must be titled differently
Authors: You were totally right. It has been changed to “2.4. Materials”.
R.2.7.- Avoide Type I error? in the scientific world of "inferential statistics" not possible
Authors: We see your point. Thus, we have moderated the statement, changing “avoid” to “minimize”.
R.2.8.- V24.0 - You need to spell out "version"
Authors: Done.
R.2.9.- missing citation and not properly referenced your source
Authors: Thank you for pointing us this typo. This reference has been included.
R.2.10.- red highlighted areas? should be avoided prior to resubmitting your manuscript
Authors: Thank you for your comment. We have corrected this issue in the style of the manuscript. As commented above, a mistake took place before the manuscript submission.
I would like to sincerely thank for your attention and time devoted to the present work, and we will be very pleased to publish it in your fine journal

Round 2
Reviewer 2 Report
Thoroughly reviewed your comments. I belive it is now ready to be piublished in this journal. Appreciate your hardwork. Have a wonderful day
This manuscript is a resubmission of an earlier submission. The following is a list of the peer review reports and author responses from that submission.
Round 1
Reviewer 1 Report
Introduction
The second paragraph, lines 40-41: Also, “he” great development…”
The third paragraph, lines 46-48: “padel has other associated risks…”. The sentence is confusing. Are the authors talking about cardiovascular complications or risk of injury?
The fourth paragraph, line 51: Is that common to abbreviate parasympathetic nervous system as SNP? It does not seem intuitive. How about PNS?
The ninth paragraph, lines 101-103: “Our study…”. Please rewrite and simplify the sentence. The sentence contains redundant phrases. A suggested sentence is “Our study seeks to be the first in the padel research to monitor HRV responses and characterize them using linear and non-linear metrics in three moments.” This simple sentence carries enough information to the readers.
Lines 104-107: “this study aimed to establish differences…” It should be “document changes” rather than “establish differences” because the authors are looking at time course of HRV over the period. Also, I don’t think this study establishes the changes in the HRV over a given timeline. The study simply documents the changes. Please delete the last aim of the study “Furthermore, this study also aimed to provide…”. It is almost redundant with the preceding aim.
Materials and Methods
The first paragraph: The authors described their study as the cross-sectional study. In the next sentence, the authors reports “the intervention was carried out…”. It doesn’t make sense. Consider to edit the sentence into “the study was carried out…”.
The fourth paragraph: the authors report “A technician randomly allocated the participants into groups of four…”. But the authors report the study consisted of a total of 27 players. So, 6 groups (4 each)? What about the rest of 3 players?
I am not very convinced of how HRV data were collected in the study. For example, the authors used a 5-minute protocol for resting HRV assessment. Did this 5-minute protocol have another 5-minute acclimatization period before the assessment (so, a total of 10 minutes: 5 minutes of acclimatization and 5 minutes of resting HRV measurement). The 10-minute protocol, which includes 5-minute HRV measurement, is probably more standard method of resting HRV measurement. I understand that 5-minute protocol is gaining popularity, but please clarify. Also, the authors report that the strap was placed on the participants’ chest area. Please describe “chest area” in detail. I assume the authors meant “under the chest muscles” or something like that, which is a standard placement of the strap. Lastly, the authors report that they used Kubios software to analyze the HR data and calculate HRV. The authors then report that they used MATLAB and Statics Toolbox. What did the authors actually do with MATLAB and toolbox? I understand that Kubios software can get researchers HRV data by itself. Did the authors use additional software to get more HRV data that Kubios cannot give? Please clarify and revise the HRV assessment.
Results
The first paragraph: The authors report “Table 2 shows the descriptive analysis for all dependent variables across the HRV and the effects performed in the different moments (baseline, game 30 min, game 60 min, 171 game 90 min, and recovery)”. This sentence needs to be re-written. A suggested sentence is “Table 2 shows the descriptive analysis for all HRV data across the time points”. This is sufficient.
The second paragraph: What is “complementary Figure 1”? and what does Figure 1 show? Please combine this paragraph with the next paragraph to describe Figure 1.
Figure 1 caption (and throughout the manuscript): Evolution? Please change it into “changes”.
Discussion
I don’t know if the current manuscript is the best way to discuss HRV data. In general, HRV analysis gives a number of HRV data (LF, HF, etc.), as seen in the current manuscript. A problem of HRV research is that all of the HRV parameters don’t have well-defined interpretation behind the parameters. For example, most time-domain and HF-HRV are known to reflect parasympathetic nervous system activity, which is closely linked to health and diseases. Compared to these parameters, most of other parameters are relatively weak to indicate physiological background; therefore, they are hard to interpret as health and disease markers. Rather than putting every HRV parameter and discussing all, I would focus on several parameters that have the established physiological background and discuss them. The authors could still present other parameters in the table since the aim of this study shows the changes in HRV over the time-course. That way, I think the discussion will generally look better.
Author Response
Dear reviewer, thank you very much for all your comments.
All comments and suggestions were taken into account.
We believe that the article had a substantial increase in its quality.
Thank you very much
Introduction
1.a) The second paragraph, lines 40-41: Also, “he” great development…”
R: Has been replaced to: “this sport's development and expand.”
1.b) The third paragraph, lines 46-48: “padel has other associated risks…”. The sentence is confusing. Are the authors talking about cardiovascular complications or risk of injury?
R: The sentence has been split into two: “However, padel has been associated with several risks, with an estimated risk of injury of 2.75 per 1000 hours of play. These injuries affect more frequently amateur players, and part of them are specific cardiovascular complications.”
1.c)The fourth paragraph, line 51: Is that common to abbreviate parasympathetic nervous system as SNP? It does not seem intuitive. How about PNS?
R: All of them were changed.
1.d)The ninth paragraph, lines 101-103: “Our study…”. Please rewrite and simplify the sentence. The sentence contains redundant phrases. A suggested sentence is “Our study seeks to be the first in the padel research to monitor HRV responses and characterize them using linear and non-linear metrics in three moments.” This simple sentence carries enough information to the readers.
R: Replaced by the suggest sentence.
1.e)Lines 104-107: “this study aimed to establish differences…” It should be “document changes” rather than “establish differences” because the authors are looking at time course of HRV over the period. Also, I don’t think this study establishes the changes in the HRV over a given timeline. The study simply documents the changes. Please delete the last aim of the study “Furthermore, this study also aimed to provide…”. It is almost redundant with the preceding aim.
R: Last sentence removed, and the second changed as suggested
Materials and Methods
1.f)The first paragraph: The authors described their study as the cross-sectional study. In the next sentence, the authors reports “the intervention was carried out…”. It doesn’t make sense. Consider to edit the sentence into “the study was carried out…”.
R: Changed as suggested
1.g) The fourth paragraph: the authors report “A technician randomly allocated the participants into groups of four…”. But the authors report the study consisted of a total of 27 players. So, 6 groups (4 each)? What about the rest of 3 players?
R: were 7 groups of 4, one of the measures did not have quality enough to be part of the sample.
1.h) I am not very convinced of how HRV data were collected in the study. For example, the authors used a 5-minute protocol for resting HRV assessment. Did this 5-minute protocol have another 5-minute acclimatization period before the assessment (so, a total of 10 minutes: 5 minutes of acclimatization and 5 minutes of resting HRV measurement). The 10-minute protocol, which includes 5-minute HRV measurement, is probably more standard method of resting HRV measurement. I understand that 5-minute protocol is gaining popularity, but please clarify.
R: Thank you for your interesting comment. We have rewritten and added more information regarding your concern. As you commented and methodological papers recommend, a prior rest (in order to ensure that it is a valid baseline). Thus, participants were at rest during 15 minutes prior to collection of baseline HRV at sitting position. We have added this relevant information to the manuscript.
1.i)Also, the authors report that the strap was placed on the participants’ chest area. Please describe “chest area” in detail. I assume the authors meant “under the chest muscles” or something like that, which is a standard placement of the strap
R: Altered.
1.j )Lastly, the authors report that they used Kubios software to analyze the HR data and calculate HRV. The authors then report that they used MATLAB and Statics Toolbox. What did the authors actually do with MATLAB and toolbox? I understand that Kubios software can get researchers HRV data by itself. Did the authors use additional software to get more HRV data that Kubios cannot give? Please clarify and revise the HRV assessment. - Kubios was only to process and obtain data, posteriorly all the treatments and data extraction were made trough MATLab.
R: Yes, the calculation of the treatment variables were obtained by Kubios and recorded in an excel file. Still, access to the data obtained by Kubios was obtained more directly and with less time consumption through Matlab. Still, no additional variable was calculated from the MatLab. We will change the text to make it more understandable to the reader.
“All HRV indices calculated by Kubios were extracted using MATLAB Release 2019a (The MathWorks, Inc., Natick, Massachusetts, United States)”
Results
1.k)The first paragraph: The authors report “Table 2 shows the descriptive analysis for all dependent variables across the HRV and the effects performed in the different moments (baseline, game 30 min, game 60 min, 171 game 90 min, and recovery)”. This sentence needs to be re-written. A suggested sentence is “Table 2 shows the descriptive analysis for all HRV data across the time points”. This is sufficient.
R: Changed as suggested.
1.l) The second paragraph: What is “complementary Figure 1”? and what does Figure 1 show? Please combine this paragraph with the next paragraph to describe Figure 1.
R: Due to the above, we performed a phrasing combination
1.m) Figure 1 caption (and throughout the manuscript): Evolution? Please change it into “changes”.
R: Replaced as suggested
Discussion
1.n) I don’t know if the current manuscript is the best way to discuss HRV data. In general, HRV analysis gives a number of HRV data (LF, HF, etc.), as seen in the current manuscript. A problem of HRV research is that all of the HRV parameters don’t have well-defined interpretation behind the parameters. For example, most time-domain and HF-HRV are known to reflect parasympathetic nervous system activity, which is closely linked to health and diseases. Compared to these parameters, most of other parameters are relatively weak to indicate physiological background; therefore, they are hard to interpret as health and disease markers. Rather than putting every HRV parameter and discussing all, I would focus on several parameters that have the established physiological background and discuss them. The authors could still present other parameters in the table since the aim of this study shows the changes in HRV over the time-course. That way, I think the discussion will generally look better.
R: Thank you for your valuable and constructive feedback. Following you suggestion, we have rewritten the discussion focusing on those parameters with enough physiological background. We believe that now the discussion is more rigorous.
Reviewer 2 Report
Thank you for your submission.
This study investigated HRV before, during, and after a game of Padel. The authors claim that no previous study has characterized HRV during padel matches. The results report a change in autonomic regulation between game and recovery time points. Although the manuscript appears to be clear, there are several suggestions that I recommend for the authors. Firstly, I think the paper should be read and edited by a native English speaker prior to resubmission.
Abstract
Introduction
can you give any size parameters on the court for the game? Is the sport somewhat similar to tennis?
lines 40-41 - "he great development...."?
how does the CV risk compare with other sports? higher/lower?
line 49 - "The HRV..." not sure you need "The"?
line 51 -"parasympathetic (SNP)"? do you mean PNS?
I think you need to create a better link between HRV and the game of padel or how measuring it can help? (reducing CV complications)
line 91 - "during padel practice"? or a game?
line 94 - FC? what does this stand for?
line 101 - not sure saying both "pioneer" and "first worldwide study" here
Methods
would fitness level alter one's HRV? was this taken into account?
table 1 - superscript 2 in kg/m2
I think you need much more detail for the HRV measurements
line 134/135 - were the moments random and different for everybody then during the time frames (e.g., 30-60 mins)?
add sub-headings in this section
Results
figure 1 - are there any y-axes titles?
Discussion
lines 207-211 - almost restating results here rather than describing and discussing
fitness level must have an impact on these HRV parameters? this is a big problem for me
I think you also need to compare and contrast what other studies have found before, during, and after sporting competition of approximately the same intensity (in relation to the HRV data)
no changes in any HRV parameter during? what does this tell us? is it similar to previous findings?
there are also no specific categories or characteristics reported?
conclusion needs to be more descriptive and have application to the sport itself - what can players, coaches, trainers take away?
Author Response
Dear reviewer, thank you very much for all your comments.
All comments and suggestions were taken into account.
We believe that the article had a substantial increase in its quality.
Thank you very much
2.a) Firstly, I think the paper should be read and edited by a native English speaker prior to resubmission.
R: All doc is changed
Abstract
Introduction
2.b)can you give any size parameters on the court for the game? Is the sport somewhat similar to tennis?
R: Added - 10 x 20 metres
2.c)lines 40-41 - "he great development...."?
R: Has been replaced to: “this sport's development and expand.”
2.d) how does the CV risk compare with other sports? higher/lower?
R: Similar to Tennis, added in the paper.
2.e) line 49 - "The HRV..." not sure you need "The"?
R: Removed as suggested
2.f) line 51 -"parasympathetic (SNP)"? do you mean PNS?
R: All of them were changed.
2.g) I think you need to create a better link between HRV and the game of padel or how measuring it can help? (reducing CV complications)
R: Throughout the review, we tried to establish a relationship, as proposed by the reviewer (introduction).
2.h) line 91 - "during padel practice"? or a game?
R: replaced to matches
2.i) line 94 - FC? what does this stand for?
R: HR, has been replaced
2.j) line 101 - not sure saying both "pioneer" and "first worldwide study" here
R: replaced by “Our study seeks to be the first in the padel research to monitor HRV responses and characterize them using linear and non-linear metrics in three moments.” as suggested by reviewer 1
Methods
2.k) would fitness level alter one's HRV? was this taken into account?
R: Regarding physical fitness level, we ensured that all the participants practiced a similar volume of padel (as we stated in the inclusion criteria). However, it is quite normal to find differences in physical fitness at amateur level. This would make more ecological and valid the obtained results. Nevertheless, as our objective was not to establish normative data, we truly believe that the aim of the study was not altered by the differences at physical fitness level. This issue has been added to the limitation section.
2.l) table 1 - superscript 2 in kg/m2
R: Altered
2.m) I think you need much more detail for the HRV measurements
R: Thank you for your recommendation. We have provided further details regarding HRV measurements.
2.n) line 134/135 - were the moments random and different for everybody then during the time frames (e.g., 30-60 mins)?
R: We studied three time intervals five-minute moments throughout the ninety minutes of the activity. The choice was made in the initial block of thirty minutes, the second five minutes between thirty and seventy minutes and finally, the last block of five minutes for analysis was obtained in the interval between sixty and ninety minutes
2.o) add sub-headings in this section:
R: added
Results
2.p) figure 1 - are there any y-axes titles?
R: The titles on the axis, we put in table 2. And we've completed the figure caption
Discussion
2.q) lines 207-211 - almost restating results here rather than describing and discussing
R: We totally agree with your appreciation. Thus, we have reduced this first paragraph.
2.r) fitness level must have an impact on these HRV parameters? this is a big problem for me
R: Thank you for your appreciation. It is true that people with higher physical fitness level report higher level of HRV. Thus, we stated as an inclusion criterion, that only those players who practice padel between four to six hours/week would be included. However, the aim of the study was not to establish normative data but reporting the evolution of HRV during exercise. Thus, we believe that the physiology would be quite similar, independently of their physical fitness. As a proof of this, you can see the SD of the different variables and you can observe that it is not present a huge variability among participants. Nevertheless, we have acknowledged this issue in the limitation section.
2.s) I think you also need to compare and contrast what other studies have found before, during, and after sporting competition of approximately the same intensity (in relation to the HRV data)
R: Thank you for your valuable feedback. Following your suggestion, we have discussed articles which analysed the HRV during and after different exercise modalities.
2.t) no changes in any HRV parameter during? what does this tell us? is it similar to previous findings?
R: We agree that this statement was not enough discussed. Thus, we have included that since changes in HRV during exercise are intensity-dependent (Michael 2017), these findings could mean that intensity is maintained (without drastic increments of intensity) during the padel game.
2.u) there are also no specific categories or characteristics reported?
R: We are not sure about the meaning of this question. Nevertheless, we have reported in the discussion that padel players which participated in our study, were classified as medium level. Unfortunately, we did not measure other level to make comparisons. And to our knowledge, this is the first study of HRV in this modality.
Conclusion needs to be more descriptive and have application to the sport itself - what can players, coaches, trainers take away?
R: Following your recommendation, we have modified the conclusion.
Round 2
Reviewer 2 Report
Thank you for your resubmission.
It is clear the manuscript has not been proofread by a native English speaker. For example, in the abstract one sentence reads "The measures differences were assessed through an ANOVA." There are also many further examples of similar poor grammar and sentence structure - "Padel game significantly
impacted the autonomic modulation in amateur players." The first sentence of the manuscript doesn't make sense "Padel is a racket sport, practiced in pairs, in a rectangular playing area of 10 per 20 metres divided into two halves by a central net." This follows suit throughout the paper and is not acceptable for this high level journal. On the grounds that the manuscript is not up to sufficient writing quality, I am recommending that the editor reject the manuscript.